# Machine Learning to Evaluate Impacts of Flood Protection in Bangladesh, 1983–2014

**Achut Manandhar** [1,*] **, Alex Fischer** [2] **, David J. Bradley** [2,3,4] **, Mashfiqus Salehin** [5] **,
M. Sirajul Islam** [6] **and Rob Hope** [2] **and David A. Clifton** [1]

[1] Department of Engineering Science, University of Oxford, Oxford OX3 7DQ, UK;
david.clifton@eng.ox.ac.uk
[2] School of Geography and the Environment & Smith School of Enterprise and the Environment,
University of Oxford, Oxford OX1 3QY, UK; alexander.fischer@smithschool.ox.ac.uk (A.F.);
david.bradley@lshtm.ac.uk (D.J.B.); robert.hope@ouce.ox.ac.uk (R.H.)
[3] Department of Zoology, University of Oxford, Oxford OX1 3SZ, UK
[4] London School of Hygiene and Tropical Medicine, University of London, London WC1E 7HT, UK
[5] Institute of Water and Flood Management (IWFM), Bangladesh University of Engineering
and Technology (BUET), Dhaka 1000, Bangladesh; mashfiqussalehin@iwfm.buet.ac.bd
[6] Laboratory Sciences and Services Division, International Centre for Diarrhoeal Disease Research,
Dhaka 1000, Bangladesh; sislam@icddrb.org
* Correspondence: achut.manandhar@eng.ox.ac.uk

**Abstract:** Impacts of climate change adaptation strategies need to be evaluated using principled methods spanning sectors and longer time frames. We propose machine-learning approaches to study the long-term impacts of flood protection in Bangladesh. Available data include socio-economic survey and events data (death, migration, etc.) from 1983–2014. These multidecadal data, rare in their extent and quality, provide a basis for using machine-learning approaches even though the data were not collected or designed to assess the impact of the flood control investments. We test whether the embankment has affected the welfare of people over time, benefiting those living inside more than those living outside. Machine-learning approaches enable learning patterns in data to help discriminate between two groups: here households living inside vs. outside. They also help identify the most informative indicators of discrimination and provide robust metrics to evaluate the quality of the model. Overall, we find no significant difference between inside/outside populations based on welfare, migration, or mortality indicators. However, we note a significant difference in inward/outward movement with respect to the embankment. While certain data gaps and spatial heterogeneity in sampled populations suggest caution in any conclusive interpretation of the flood protection infrastructure, we do not see higher benefits accruing to those living with higher levels of protection. This has implications for Bangladesh's planning for future and more extreme climate futures, including the national Delta Plan, and global investments in climate resilient infrastructure to create positive social impacts.

**Keywords:** Bangladesh; climate resilience; flood protection; machine learning; socio-environmental impacts

## 1. Introduction

Climate change is expected to increase the frequency and extent of extreme flood events, which will directly impact the environment and the livelihoods of people in the affected areas [1,2]. Low-lying coastal regions of the world are particularly vulnerable to these flood events and sea level rise [3,4]. The issue is compounded in countries such as Bangladesh, where about 60% of the country is lower than 6 m above the sea level [5] while more than 70% of land is used for agriculture, the country's primary

economic source [6]. Although there have been global investments on climate change adaptation, these adaptation measures can have both beneficial and unintended detrimental consequences when wider issues or longer time frames are considered [7,8]. Integration of adaptation actions and policies across sectors at different spatio-temporal and societal scales remains a key challenge to achieve effective adaptation in practice [9]. Another major challenge is the lack of consistent empirical methods linking climate change to the impacts on the environment and the livelihoods of people [10]. Although success of these interventions relies on developing principled methods to monitor and evaluate the impacts across different environmental and socio-economic factors [11], currently there is a lack of such methods in the literature. This work aims to bridge a key gap in knowledge by proposing rigorous analytical methods to evaluate the impacts of adaptation measures.

Bangladesh has had five decades of political and policy attention focused on implementing flood mitigation strategies, whose evaluation has been documented in the past literature on Bangladesh flood protection infrastructure. The total flood protection coverage area currently stands at 5.37 million ha, more than one third of the country [12,13]. Primarily aimed to protect monsoon crops and prevent damage to homesteads, these interventions have often not considered the social, economic, and environmental dimensions of water resource management [13]. While the interventions led to several positive impacts, they also resulted in considerable medium to long-term negative consequences in many places. The flood secure environment facilitated enhanced economic activities in the protected areas, e.g., increased agricultural output [14]. But when embankments failed to provide protection during moderate to extreme floods, the resulting damage was higher owing to the accelerated economic activities compared to areas outside embankment [13,15]. Floodplains were deprived of several environmental and ecological functions, including improvement of soil fertility from silt-laden inundation water, groundwater recharge, and biodiversity. The disruption in hydraulic connection between river and floodplain led to substantial damage to fisheries and local boat transports [14–18], compromising the livelihood activities of people, especially the marginalized.

Previous evaluations of flood protection investments in Bangladesh have widely suggested that it has been difficult to attain the stated objectives of the interventions based on only technical and economic viability, but without giving due consideration to the hydromorphological features of the floodplain and the socio-economic condition of its inhabitants. Despite the negative consequences of embankments, the popular demand for flood protection by people has been high. Hence, the priority of the Government started to shift from traditional flood control to flood management towards the later stages of the Flood Action Plan (FAP) in late 1980s. Here, flood control refers to the conventional method of constructing an embankment and drainage regulator whereas flood management refers to mitigating flood damage without causing degradation of the floodplain environment, which might involve implementing floodplain land use regulation that identifies floodplain zones and enforces appropriate planning and design during construction of infrastructures in these floodplain zones to account for flood and preservation of floodplain resources and environment. There was a real paradigm shift to integrated flood management, i.e., covering issues relevant to not only flood but also drainage, irrigation, navigation, environment and socio-economic development, which was subsequently reflected in the National Water Policy (NWP) in 1999 [19] and the National Water Management Plan (NWMP) [12]. A combination of structural and non-structural measures was envisioned, with full structural protection against floods in regions of economic importance (such as metropolitan areas), a reasonable degree of protection in other critical areas (such as district towns), and flood proofing measures in the rural areas. However, translating integrated management into action, particularly at the program and project levels, has remained a major challenge [20].

The Bangladesh Delta Plan 2100 also gives more emphasis to restoration, redesign and modification of existing embankments and associated structures, and a high importance to the urgency of maintenance [21]. New developments are envisioned only for protecting economic strongholds and critical infrastructure. Most are already in place but requires improvement; additional developments will be required in some areas. In order to properly evaluate the impacts of these investments, this work

strongly recommends adoption of principled analytical methods early on, considering socio-economy and environment across longer time frames, so that appropriate studies are conducted at the outset, e.g., baseline and periodic survey and monitoring to assess the impact of interventions.

The changes in Bangladesh Delta are affected by many factors, upstream interventions being one of them. In this work, we evaluate the impact of flood infrastructure on a large project in Bangladesh. We took advantage of an existing data set of this project to further clarify evaluation outcomes in Bangladesh flood management using machine learning approaches. We chose the site also because it had robust historic continuous data to trial machine learning that enabled the type of analysis that other sites could not. Machine learning, generally considered a subset of artificial intelligence, is a field of study that uses algorithms and statistical models to learn patterns from data so that useful inference may be made about new data. Machine-learning approaches are useful in the context of this study because they provide robust metrics to evaluate the impacts of interventions as well as identifying the most informative indicators.

*Context and Related Works*

The Meghna–Dhonagoda Irrigation Project (MDIP) is one of the largest flood control, drainage and irrigation projects in Bangladesh, implemented in 1988 with the objective to protect the area in Matlab North from river flooding and drainage congestion during the monsoon via embankment and regulators. The primary aim was to improve agricultural conditions in monsoon, with special reference to encouraging introduction to high yielding variety (HYV) monsoon rice (Aman), and also to provide irrigation from surface water in the Rabi (dry) season and early monsoon seasons. Located 55 km south-east of the capital city Dhaka, the study area is 184.4 km$^2$ with a population of 230,185 as of 30 June 2014 [22]. The Dhonagoda river bisects the area into Matlab North and Matlab South. The embankment protects the Matlab North area from flooding from Meghna river on the north and west and Dhonagoda River on the east and south (Figure 1).

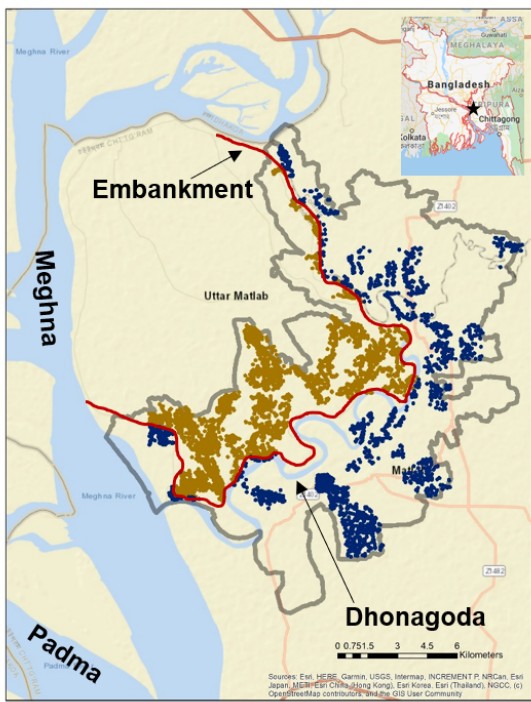

**Figure 1.** Baris (cluster of households) are colored based on whether they are located inside (brown) or outside (blue) the embankment (red).

This study area provides a unique setting to evaluate the long-term impacts of an embankment on households living on both sides of the embankment by using an unrelated but population-wide Health and Demographic Surveillance System (HDSS) [22,23] managed by the International Centre on Diarrheal Disease Research, Bangladesh (icddr,b). The research site was created in 1963, over 25 years before the embankment's construction, to provide field-based research on cholera vaccines and treatments. The site maintains a primary research focus on studying public health interventions and demographic changes. The study site evolved around two sets of comparison zones. The primary division is a quasi-experimental maternal and child health and family planning program where half the population received reproductive health interventions and home visits from community health workers, while the other half received the government's public health services [22].

The other key division across the study area is between those within and those outside the embankment. The embankment-led agricultural and related economic variables are not rigorously tracked through the icddr,b data systems and health-focused research agenda [24]. Discussion of the embankment's impact did not appear in the icddr,b annual reports until the 2014 analysis of the socio-economic survey and is not a feature of these reports which serve as a core site output [23]. However, the embankment provided a clear division of the population between people living inside vs. outside the embankment [22,25]. The HDSS data, spanning pre–post-embankment periods provide a unique opportunity to study the long-term impact of embankment.

Multiple studies have used the HDSS data to advance analysis around the indirect impacts of embankment [25–30]. Earlier studies of child mortality in Matlab in 1981 identified the socio-economic factors as indicators of severe malnourishment, impacting child mortality, which was a basis for socio-economic survey focus on household assets, education and occupation [26]. A later study during 1983–1992 showed that child mortality was higher outside than inside the embankment, with the differences particularly significant for deaths caused by infectious diseases [27]. The study recommended comparison of long-term mortality data with migration patterns and other factors (e.g., proximity to main rivers). Another study of the embankment's impact on cholera during 1983–2003 showed that after controlling all other environmental variables, living inside the embankment area does not appear to affect whether the household experiences cholera. Counter-intuitively, among households where cholera is reported, living inside the embankment area significantly increases the number of cholera cases, possibly due to a combination of environmental factors and behavioral change [29]. In particular, a joint study by Bangladesh Rural Advancement Committee (BRAC) and International Centre for Diarrheal Diseases Research, Bangladesh (icddr,b) analyzed the impact of embankment on both environment and people [28]. Based on both quantitative and qualitative surveys conducted in 1992 and 1996 respectively, the study revealed both positive and negative impacts. The positive impacts were associated with a higher level of agricultural yields and economic prosperity, particularly for a subset of farmers, while the negative impacts were associated with lower fish catch and intake of fruit and vegetables, displacement of poorer households, and complaints about ill health due to greater demand for agricultural labor [28]. However, the core findings around welfare differences were inconclusive. The study recommended doing a follow-up study using longer-term data to analyze how these impacts would evolve over time. Another study found that the embankment resulted in a net welfare loss, which had been an outcome of higher than anticipated construction costs, lower benefits to agriculture due to loss of soil fertility over time, higher waterlogging damages and the highly negative impact on capture fisheries [30]. The embankment also caused negative distributional outcome, with big landowners benefiting from increased agricultural production, reduced property damage, and increased livestock and aquaculture production. In contrast, the traditional fishermen and the river transport workers, belonging to the poor sections of the community, suffered significantly negative impact. However, the variables related to these impacts were not monitored consistently over time, limiting ability to include in longer-term impact assessments.

Most studies occurred within a decade of the embankment completion, showing only the short-term impacts. There is a limited body of literature exploring long-term impacts outside agricultural productivity, providing a core motivation of this paper to re-purpose multi-decadal HDSS data. Using this data, machine-learning approaches can be used to empirically evaluate the impact of embankment and identify the most informative socio-economic indicators. Besides, these approaches provide robust metrics to evaluate the model's accuracy and generalizability to future examples. We explore the extent to which machine learning can provide analytical insights from detailed historical data on long-term impacts from climate resilient infrastructure to help guide future policy and investments.

## 2. Methods

### 2.1. Machine-Learning Approaches

Machine-learning approaches have found useful applications in a wide variety of fields, including text processing, computer vision, healthcare, finance, and robotics [31–36]. Recently, these approaches have also been applied to socio-economic [37–40] and environmental [41–45] studies.

We implement machine learning on two types of data to answer two specific questions:

1. Based on multidecadal socio-economic survey data, are there any significant differences in socio-economic status of households living inside vs. outside embankment over time, and which variables are most informative of the differences?
2. Based on multidecadal events data, are there significant differences in mortality and migration patterns of households living inside vs. outside embankment over time?

To answer the first question, socio-economic variables corresponding to a household (inputs) are mapped to a binary label (output). The responses provided by households during a socio-economic survey are used to determine whether the households live inside or outside embankment. By comparing classification outputs over time, we can infer whether the embankment has caused differences in welfare inside vs. outside embankment over time. To answer the second question, a regression model is used to map time (input) to event rate (output), e.g., mortality rate per year. After learning two independent regression models for two event rates corresponding to inside vs. outside embankment, we can analyze whether the embankment has caused differences in the event rates over time. An array of approaches exists to perform classification and regression. The ones adopted in this work are motivated by their suitability to model the available data in answering the aforementioned questions.

#### 2.1.1. Classification Approaches

For each household, its label $y_n$ (output), indicating whether it lies inside or outside embankment, is defined in terms of its responses to $D$ survey questions $\mathbf{x_n} = [x_{n1}, ..., x_{nD}]^T$ (inputs) as follows:

$$y_n = f(\mathbf{x_n}) + \epsilon_n, \tag{1}$$

where $\epsilon_n$ is the residual or noise, and $f()$ is the mapping function specific to the type of classifier. The mapping function and its parameters can be learned using a collection of household survey responses and their corresponding labels (termed "training" examples). Once a mapping function is learned, it can be used to predict labels for new previously unseen (termed "test") examples. By evaluating whether the labels are correctly predicted for test examples, we can determine the ability of a classifier in discriminating two classes. The simplest mapping function is a linear function of the inputs as follows

$$y_n = w_0 + w_1 x_{n1} + ... + w_D x_{nD}, \tag{2}$$

where the weights $\mathbf{w} = [w_0, ..., w_D]^T$ are known as the parameters of the mapping function. However, this simple linear discriminant function [31] cannot model interactions between variables or other

complex phenomena. Generally, classification techniques learn a non-trivial mapping function, capable of performing nonlinear classification, which are more relevant to model real-world examples.

Some standard classification approaches are Logistic Regression (LR) [31], kernel-based methods, e.g., Support Vector Machines (SVMs) [46], decision trees, e.g., Random Forest (RF) [47], and Neural Networks (NN), e.g., Stacked Auto-Encoders (SAE) [36]. Logistic Regression (LR) is one of the most popular classification approaches, which uses a logistic sigmoid function to perform probabilistic binary classification. However, without using kernels, LR is often only suitable to classify linearly separable examples. In general, we expect other above-mentioned approaches to perform as well as or better than LR.

Random Forest (RF) is a type of ensemble-based method that performs an average over an ensemble of many estimates obtained over bootstrapped subsets of data, where the ensemble of estimators can be thought of as leaves of a tree [47]. An added advantage of RF is that it also ranks the input variables by their importance to discriminate the two classes. In the context of this study, the variable importance ranking helps identify which survey questions are most informative of households living inside vs. outside embankment. When using RF, it is important to choose an appropriate number of estimators, and optimize other parameters.

Stacked Auto-Encoders (SAE) is a type of deep learning methods that first learns the lower-dimensional representation of data by constraining the hidden layers to capture the most relevant aspects of the data. Then the whole network is discriminatively fine-tuned like a feedforward neural network to perform classification [36]. Similar to Principal Component Analysis (PCA) [34], which is the most common dimension reduction technique, SAE can be used to learn useful lower-dimensional representations of data to uncover patterns in data, e.g., identify clusters of households with similar features. Unlike PCA, SAE is not limited by the Gaussian distribution assumptions of the input space, it is more flexible to model mixed data types (binary, categorical, and continuous), and being a deep learning approach, it is capable of handling large amounts of data. These are all favorable properties for analysis of socio-economic survey data. It is important to optimize SAE's parameters such as learning rate, batch size, number of epochs, number of hidden nodes, number of hidden layers, etc.

We do not go into details regarding the specifics of these approaches other than provide a general intuition for the ones that are implemented. Having experimented with a variety of these approaches, we report majority of our results based on RF, while comparing RF to LR and SAE on one specific example for reference.

### 2.1.2. Regression Approaches

Shifting from discrete to continuous output labels, for each year $x_n$ (input), the event rate for that year, i.e., its label $y_n$ (output), can be defined with (1), where $y_n$ is now a continuous variable. Different regression approaches exist, e.g., Relevance Vector Machines (RVMs) [48], Gaussian Processes (GPs) [49]. Gaussian Processes (GPs) are probabilistic methods (i.e., capable of giving uncertainty of model's predictions) that can be used to model time-series data as a distribution over function [49]. We use GPs because in addition to modeling event rates as time-series, they are also useful to compare differences in the resulting time-series models. In a Bayesian framework, the event rates $\mathbf{y} = [y_1, ..., y_N]$ for N years $\mathbf{x} = [x_1, ..., x_N]$ can be defined by a GP prior as follows

$$f(\mathbf{x}) \sim GP(m(\mathbf{x}), k(x, x^T)), \tag{3}$$

where $\mathbf{y} = f(\mathbf{x}) + \epsilon$, $\epsilon$ is the noise, $m(\mathbf{x})$ is the mean function and $k(x, x^T)$ is the kernel or covariance function. This allows the posterior distribution of the function evaluated at a finite set of points $\mathbf{x}_*$ to be a multivariate Gaussian distribution as follows

$$p(\mathbf{f}_* | \mathbf{x}_*, \mathbf{x}, \mathbf{y}) = N(\mathbf{f}_* | \mu_*, \mathbf{\Sigma}_*), \tag{4}$$

where $\mu_*$ and $\Sigma_*$ are the posterior mean and covariance, respectively. This posterior formulation allows us to compare the differences in two event rates in a principled manner, which is described in the next section. When using GPs, it is important to choose an appropriate kernel function and optimize its parameters.

*2.2. Evaluation Metric*

We evaluate inside/outside embankment classification using Receiver Operating Characteristics (ROC) curve based on k-fold (k=3) cross-validation [31], which prevents a classifier from overfitting to training examples, thus ensuring the results are generalizable. Intuitively, a diagonal ROC curve means the predictors (socio-economic survey variables) are not indicative of inside/outside class discrimination. On the other hand, the more the curve pushes towards the top-left corner, the more discriminatory are the predictors. A statistical significance test can be performed to compare the area under ROC curves (AUCs) of the later three years with the AUC of 1982 [50]. Whenever a comparison is statistically significantly different with *p*-value less than 0.01, the corresponding *p*-value is highlighted with a ∗ symbol.

In line with the previous studies [25,29], to compare differences in events data pre- vs. post-embankment, we divided the 32-year study duration into pre (1983–1989) vs. post (1990–2014) embankment periods. Within each of these two periods, we can compare the temporal differences in these events using a Gaussian Process-based Bayesian statistical significance test [51]. The posterior formulation in (4) allows us to model the differences in event rates, $\Delta \mathbf{f}_* = \mathbf{f}_{1*} - \mathbf{f}_{2*}$, as another multivariate Gaussian with mean $\Delta \mu_* = \mu_{1*} - \mu_{2*}$, and covariance $\Delta \Sigma_* = \Sigma_{1*} + \Sigma_{2*}$. Then, we say the two event rates are equal with posterior probability $1 - \alpha$ if the credible region for $\Delta \mathbf{f}_*$ includes the zero vector or, in other words, if

$$(\Delta \mu_*)^T (\Delta \Sigma_*)^{-1} \Delta \mu_* \leq \chi_\nu^2 (1 - \alpha), \tag{5}$$

where $\chi_\nu^2(1 - \alpha)$ is the $1 - \alpha$-quantile of a Chi-squared distribution with $\nu$ degrees of freedoms and $\nu$ is the number of positive eigenvalues of $\Delta \Sigma_*$ [51]. This test can be used to compare two event rates within each (pre or post) embankment period.

## 3. Data

Part of the HDSS dataset is available for this study, covering roughly one third of the study area geographically.

*3.1. Socio-Economic Survey Data*

Socio-economic survey data are available for four years, roughly a decade apart. The number of questions in surveys increased over the years from 21, 40, 62, to 146 in 1982, 1996, 2005, and 2014, covering 14791, 19448, 22799, and 25840 households, respectively. Considering 1982 as a baseline pre-embankment period resulted in only four variables equivalent across all four years. Those variables were agricultural land ownership, primary drinking water source, number of cow/buffaloes/goats owned, and boat ownership. Considering only the later three years (1996, 2005, and 2014) resulted in slightly more (12) variables equivalent across all three years. Those variables were agricultural land ownership, homestead land ownership, primary drinking water source, number of cow/buffaloes/goats owned, boat ownership, household assets—sofa, chair/table, showcase, radio, TV, bike/bicycle, primary rood structure, and sanitation facility type. For a consistent comparison, we only consider households that are common across all years, i.e., 10563 and 14276 households common across all four and the later three years, respectively. We note that the results are similar when all households per year are used. For a separate analysis specific to socio-economic survey data from 2014, whose aim was not to compare with prior years, we use all available variables and households.

## 3.2. Events Data

Events data, collected monthly to bi-monthly, correspond to birth, marriage, divorce, death, inside/outside migration, and inward/outward movement [23]. Here, *migration* refers to the migration from anywhere outside the study area into either the embanked part or the outside part of the study area; whereas *movement* refers to the movement of the study area inhabitants either into or out of the embanked area. We hypothesize that if the flood-protected area were to provide increased socio-economic benefits and stability, people would more likely move to inside the flood-protected area. Although experiments were performed with all data, we only provide results from analyzing death, internal movement, and migration data, which are presumed to be the most informative of the effect of embankment. The event counts in each group (inside/outside) are normalized by the inside/outside annual mid-year population. Since the mid-year population was unavailable for one particular year, we only use data from the remaining 31 years.

## 4. Results

### 4.1. Socio-Economic Survey Data Analysis

ROC comparison across four years in Figure 2a seems to suggest there are differences in inside/outside discrimination over time based on socio-economic variables. However, one of these variables, the ownership of a boat, is mostly a consequence of differences in habitat due to embankment rather than an indication of welfare. Indeed, when we remove the ownership of boat variable, Figure 2b shows that the discrimination does not change much over time except for 1996. To understand the increased discrimination in 1996, we rank the three variables (agricultural land ownership, primary drinking water source, and number of cow/buffaloes/goats owned) by their importance in Figure 2c. We observe that agricultural land ownership is the most discriminative variable across all four years. When we classify households based on only agricultural land ownership (results not shown), we observe that although the discrimination increases temporarily during 1996, it falls back to pre-embankment period over time. The statistical significance test in Table 1 supports that only the 1996 AUC metric is significantly different compared to 1982. The results suggest that although there was a short-term increase in agricultural land ownership for inside residents, the difference seems to have evened out over time. Despite the statistical significance test, it should be noted that all AUCs, including 1996, have low values, implying that the equivalent variables are not very discriminative. The results also highlight the need for more data on agricultural productivity and markets to further investigate the differential impact over time.

Setting aside the temporal comparison for a moment, if we were to make use of more variables for a particular year, we could learn a better model and use the classifier's outputs to further investigate the most informative variables in details. Figure 2d shows that when 122 relevant out of 146 variables from year 2014 are used, the classifier performs strong inside/outside discrimination. Figure 2e shows the top 10 informative variables, ordered by their importance—homestead land size, number of households sharing drinking water source, number of chicken/ducks, agricultural land size, fuel source is wood/gas, primary income source, primary drinking water source is Arsenic-safe tubewell/pipe, and if zakat received. Each of these variables can be visualized by aggregating the corresponding household responses at bari-level and comparing each plot to inside/outside locations of baris in Figure 1 as a reference. Figure 3i,ii show that indeed households inside the embankment appear to own more homestead land and owning homestead land appears to be correlated with owning agricultural land. Consequently, land ownership is identified as the most important discriminative indicator. However, for several other variables, the difference appears to be a consequence of the households' proximity to significant infrastructures (e.g., hospital, gasline, pipeline, deep tubewells) instead of a direct consequence of the households living inside/outside embankment. Availability of gasline (Figure 3iv) in certain parts of the study area is complementary to those households not using the more ubiquitous wood as fuel source (Figure 3iii). Likewise, availability of pipeline (Figure 3vi) is

complementary to those households not using the more ubiquitous tubewell as drinking water source (Figure 3v). Similarly, larger number of households sharing a drinking water source (Figure 3viii) appears to be complementary to those households using deep tubewells (Figure 3vii).

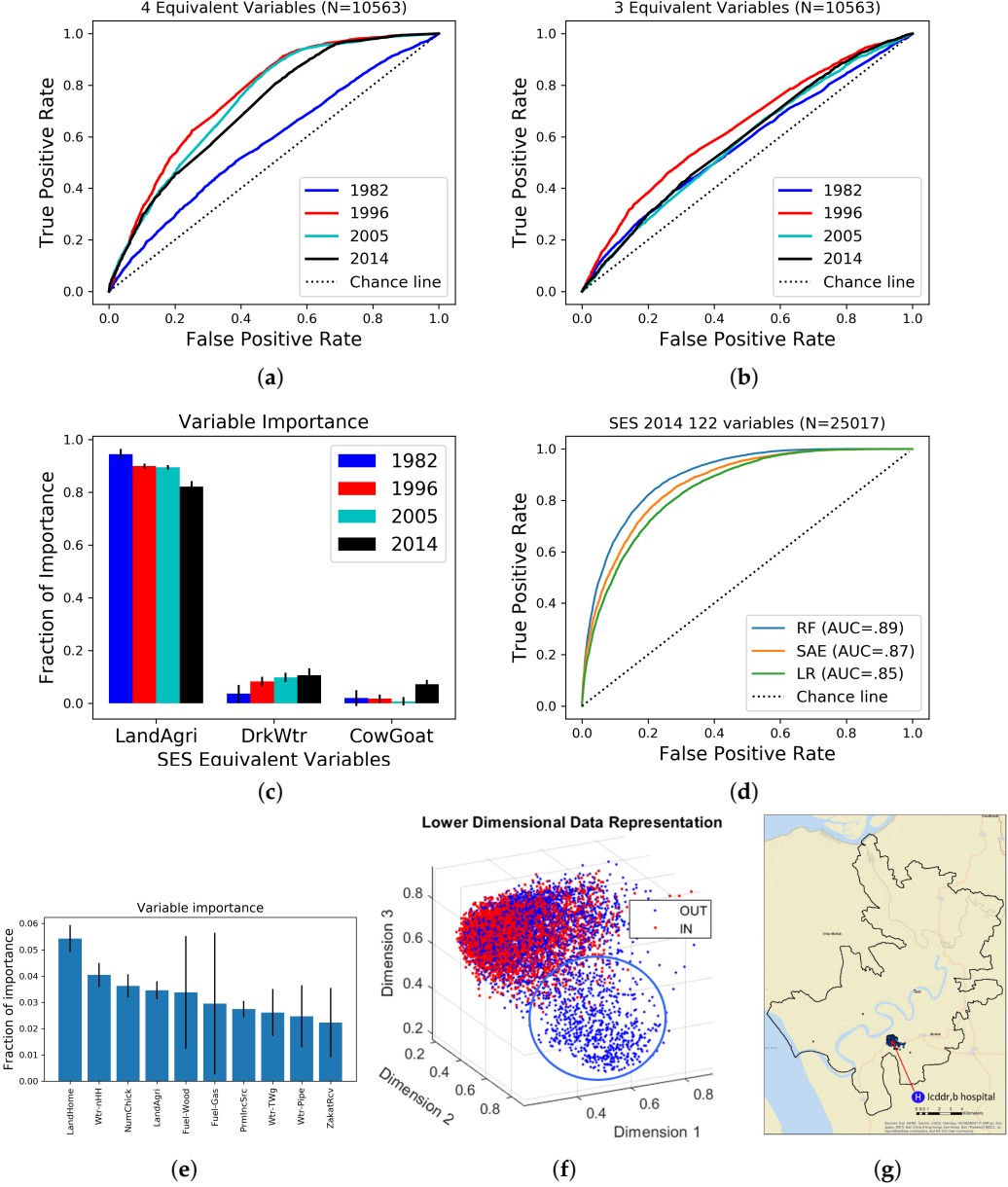

**Figure 2.** Inside vs. outside embankment classification outputs based on socio-economic survey data using (**a**) four equivalent variables across four years, (**b**) same as (**a**) excluding boat ownership variable, (**d**) 122 variables from 2014 survey, (**c**) top three, and (**e**) top ten discriminating variables identified by Random Forest classifier in respectively (**b**) and (**d**). (**f**) lower-dimensional representation of SES 2014 data, and (**g**) the location of households, corresponding to the cluster inside the blue circle, in the study area.

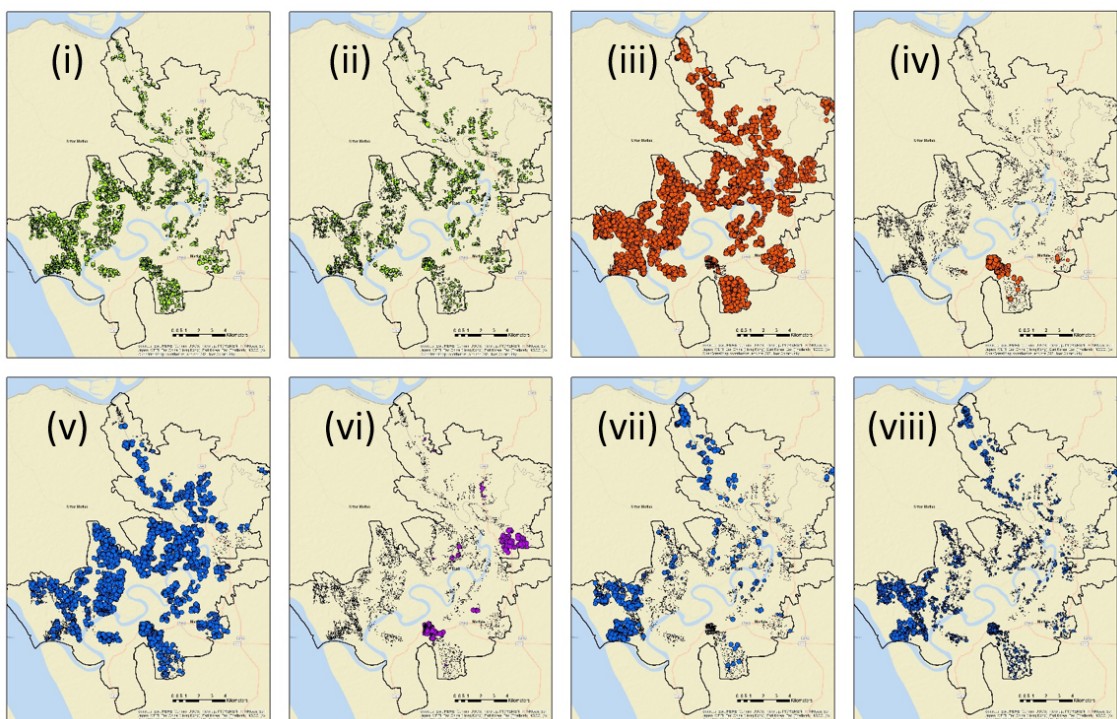

**Figure 3.** Selected variables from socio-economic survey data. Responses aggregated over bari. (**i,ii**) Homestead vs. agricultural land owned, (**iii,iv**) primary fuel source is wood/wood dust/paddy husk vs. gasline, (**v–vii**) primary drinking water source is green shallow tubewell, pipeline, vs. deep tubewell, and (**viii**) number of households sharing drinking water source.

**Table 1.** Comparison of AUCs across four years based on Wilcoxon statistic. The ∗ symbol denotes the AUCs are significantly different with $p$-value less than 0.01.

| Year | AUC | SE | $|$Year$-$1982$|$ | SE of Diff | Z Score | $p$ Value |
|------|-----|----|----|----|----|----|
| 1982 | 0.564 | 0.0056 | | | | |
| 1996 | 0.616 | 0.0055 | 0.052 | 0.0078 | $-6.6611$ | $< 0.01^*$ |
| 2005 | 0.567 | 0.0056 | 0.003 | 0.0079 | $-0.3803$ | 0.7 |
| 2014 | 0.557 | 0.0056 | 0.007 | 0.0079 | 0.8862 | 0.38 |

Referring back to Figure 2d, we note that we expect Stacked Auto-Encoders to perform as well or better than Random Forest when Stacked Auto-Encoders' parameters are exhaustively optimized. Rather than obtaining the best discrimination, the primary goal of using Stacked Auto-Encoders was to show its value in learning useful lower-dimensional representation of data. Figure 2f shows a 3-dimensional representation of 122 survey questions, which identifies a cluster of outside residents who are very different from inside residents based on their responses to the survey questions. A geospatial plot (Figure 2g) reveals that this cluster of outside households, in fact, are all located close to the icddr,b hospital, which incidentally, is associated with factors such as availability of gasline. These results suggest that although there was a significant relationship between introduction of embankment and change in land ownership patterns, recent development progress is linked to other variables (e.g., electricity/energy and proximity to services). Spatial analysis of these variables might provide useful insight to track differential progress in the study area.

*4.2. Events Data Analysis*

Figure 4 summarizes the results of events data analysis for three cases—inward/outward movement, inside/outside in-migration, and inside/outside mortality over time. The first column shows the individual event rate data modeled by independent Gaussian Processes. For each of these

plots, the dots represent the normalized event rates per year, the solid lines represent the estimated means, and the shaded regions represent the 99% confidence interval. For each of these three cases, the second column shows the estimated differences in two event rates. The more the two event rates differ, the more the curve deviates from the zero-line, represented by the dotted black line. Finally, the dashed red line marks the pre- vs. post-embankment periods. For each of the second-row plots, within each embankment period, the confidence interval region helps us visually determine if the difference in event rates is statistically significant. If the confidence interval excludes the zero-line, the difference in event rates in this period is statistically significant. These test results are also summarized in Table 2.

Figure 4a,b show that the inward movement rate increased leading up to the embankment's construction, peaking at 1988, and continued to remain significantly higher than the outward movement rate. In recent years, the difference seems to have evened out. Figure 4c,d show that after embankment's construction, the in-migration rate to the area outside of embankment has been increasing compared to inside. However, the difference in in-migration rates is not statistically significant. Based on available data, it is unclear if the decreasing inward movement and the increasing in-migration inside the embankment area are due to growing competition for land inside the embankment or due to falling motivation to move inside. In contrast, Figure 4f shows that the embankment has not caused differences in mortality inside vs. outside over time. The mortality data was disaggregated into vulnerable (under 5-year-olds or over 70-year-olds) vs. non-vulnerable population, and male vs. female. The results (not shown) are similar to Figure 4e,f. Similar analysis done using out-migration data (not shown) shows no difference inside vs. outside. These results suggest that apart from internal movement within the study area, significant differences are not observed in migration and mortality patterns inside vs outside over time.

### 4.3. Hydro-Climatic Data Analysis

To fully understand the impact of embankment, it is important to link the socio-economic impacts with the hydro-climatic events. With about 47% of the land being low lying, the Meghna–Dhonagoda project area within Matlab North used to be regularly flooded in the monsoon up to a depth of 2–3 m in the pre-project condition [52] (Saleh et al. 2000) (Figure 5b). In contrast, there is a lower proportion of low-lying land area in Matlab South (Figure 5b). While the low-lying areas get inundated during an average annual flood, the relatively higher lands are inundated only during moderate to extreme floods. Since Matlab South is only exposed to the Dhonagoda river, the area is relatively less vulnerable compared to other areas which are directly affected by floodwater from the large Meghna River.

Figures 5c,d show the inundated areas in Matlab in two major flood years, delineated from LANDSAT 4-5 TM images using Normalized Difference Water Index (NDWI) [53]. Although designed based on 1 in 100-year flood level, the embankment, suffering major breaches, could not protect the project area during the 1988 flood (corresponding to a 30-year flood). After repairs, the embankment successfully withstood floods during subsequent years, but frequently suffered many problems. Although the river water level was higher than the design 100-year flood level during the 1998 flood, the severest in recent history both in terms of magnitude and duration, the embankment was able to protect the project area from inundation (Figure 5d) (Saleh et al. 2002; [52]). The flood did; however, cause substantial damages to the embankment. No inundation due to riverine flood inside the embankment has been reported in later years. However, some areas inside the embankment are subject to waterlogging induced by rainfall [30,54]. Two pumping stations inside the project area are operated during the monsoon to drain out the accumulated rainwater.

Figure 5b shows that the inward movement rate behaves like an impulse function during the embankment's construction, propelling a substantial inward movement. However, subsequent spikes in the inward movement rates and the overall trend do not appear to be correlated with the monthly tidal data observed at a monitoring site. Further analysis of relevant data (e.g., financial

loss due to floods) may allow us to study the relationship between hydro-climatic events and socio-economic behavior.

**Table 2.** Comparison of event rates within each (pre/post) embankment period based on GP-based. Here, $\chi^2(\nu = 7, p = 0.01) = 1.24$, and $\chi^2(\nu = 24, p = 0.01) = 10.86$. The $*$ symbol denotes the event rates are significantly different with *p*-value less than 0.01.

| Event | Pre-Embankment $\chi^2(x, \nu = 7)$ | *p* Value | Post-Embankment $\chi^2(x, \nu = 24)$ | *p* Value |
|---|---|---|---|---|
| Internal Movement | 10.63 | 0.31 | 102.57 | <0.01 * |
| In-Migration | 1.82 | 1.94 | 14.19 | 1.88 |
| Mortality | 2.88 | 1.79 | 17.43 | 1.66 |

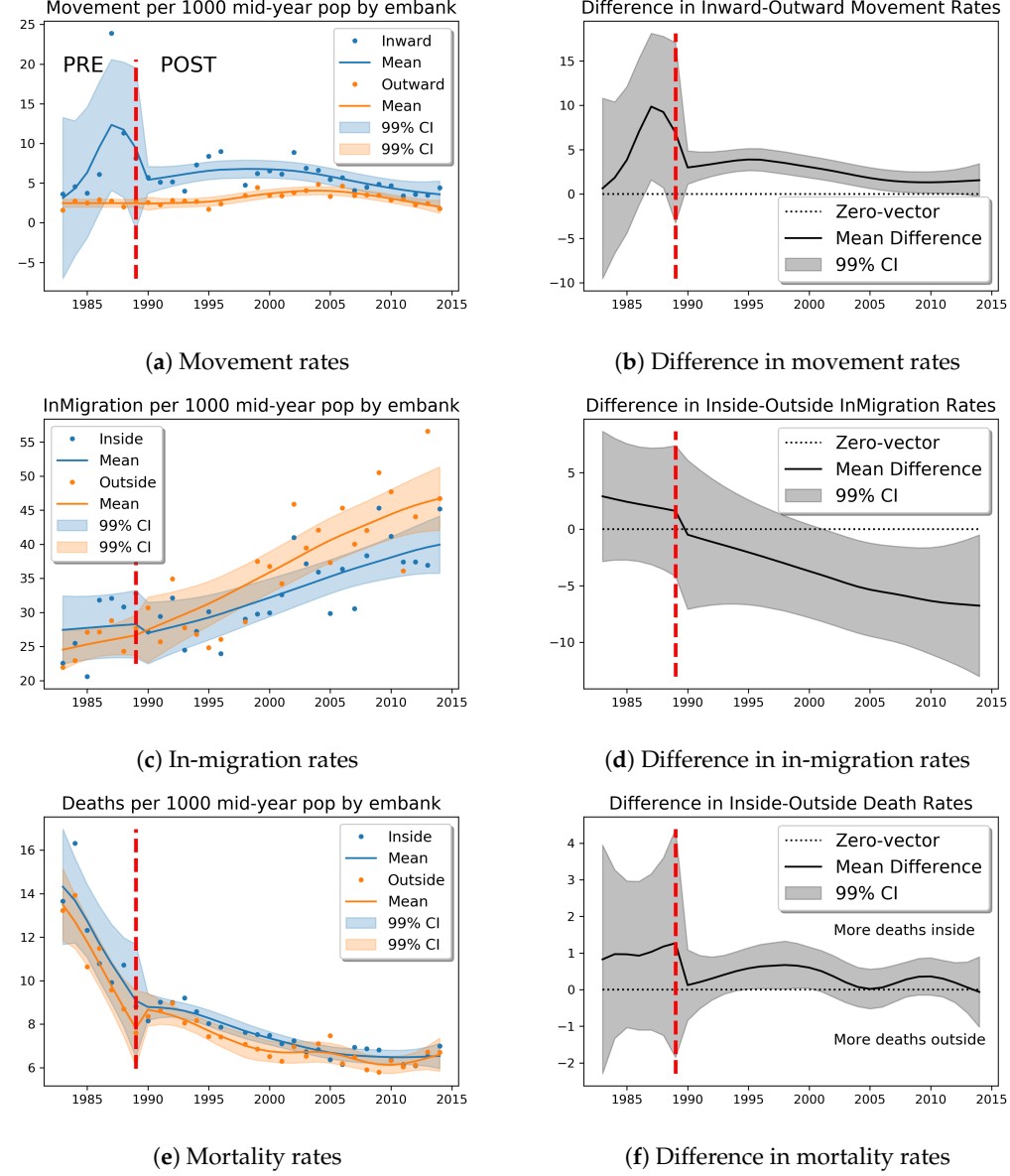

(**a**) Movement rates

(**b**) Difference in movement rates

(**c**) In-migration rates

(**d**) Difference in in-migration rates

(**e**) Mortality rates

(**f**) Difference in mortality rates

**Figure 4.** Analysis of temporal difference for three specific events— (**a**,**b**) inward/outward movement, (**c**,**d**) inside/outside in-migration, and (**e**,**f**) inside/outside mortality during 1983–2014. For each case, the left column shows the individual time-series modeled by GP, and the right column shows the estimated differences in two time-series. The red dashed line marks the pre- and post-embankment periods.

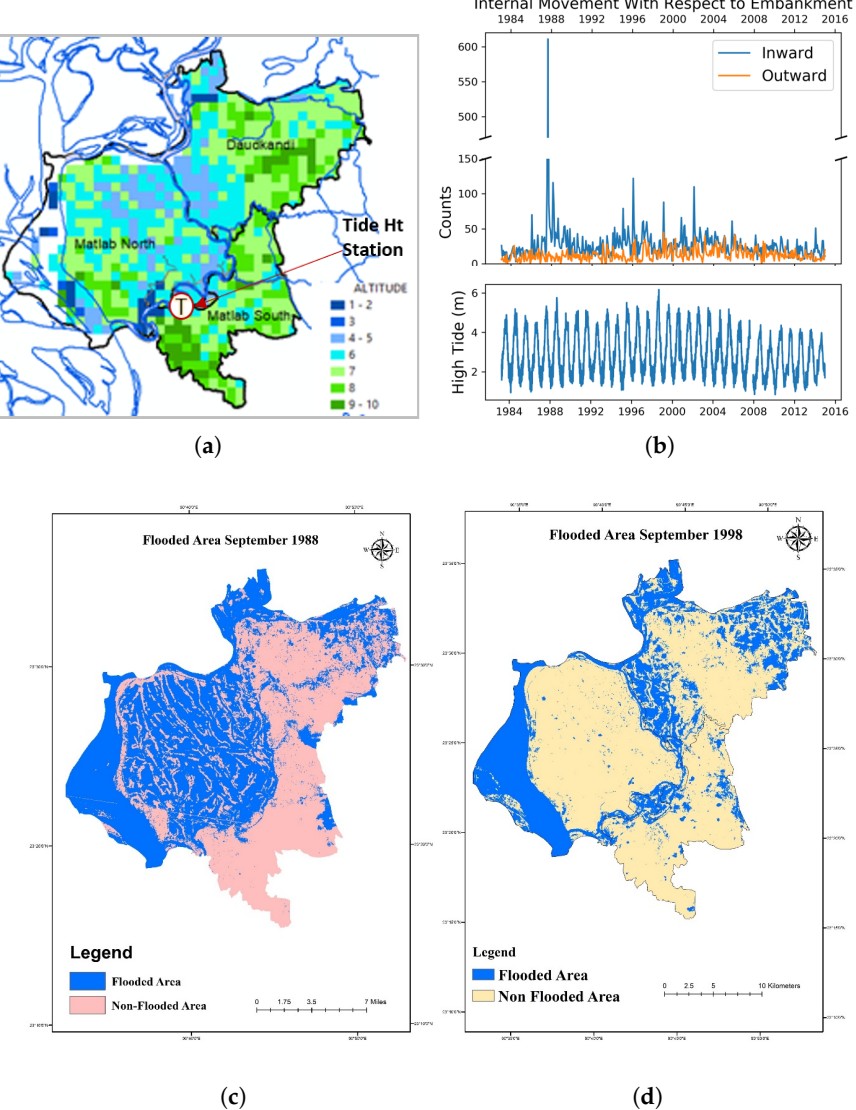

**Figure 5.** (**a**) Land elevation map of Matlab North, Matlab South and Daudkandi Upazilas based on 20 m × 200 m National DEM data (Source of data: WARPO), (**b,c**) inundated area in Matlab North, Matlab South and Daudkandi Upazilas in two major flood years [delineated from LANDSAT 4-5 TM images using Normalized Difference Water Index (NDWI)], and (**d**) (top) monthly inward/outward movement of households within Matlab, and (bottom) monthly high-tide observed at station located in (**a**).

## 5. Discussion

The study has limitations that are largely due to either unavailability of relevant data or some inconsistency in the data collected over multiple decades. Data from only one third (geographically) of the study area were available for analysis. The study may not generalize due to small study size and differences in hydroclimatic conditions. The results based on the classification approaches should be accepted with caution due to lack of enough common indicators across different years, especially the pre-embankment baseline year. We acknowledge other infrastructures (health, roads, electricity, education, etc.) have impacts in relation to the embankment. The spatial plots of a selection of socio-economic variables suggest recent social developments may be tied to these and other interventions. Similarly, data on agricultural yields, fishing, and other water-dependent economic activities would provide vital information. The limitations in data highlight the challenges of evaluating

long-term impacts of interventions such as the embankment, where relevant indicators may not be identified during the baseline study (if performed), and may not be consistently monitored over time.

Although other health event outcome related data were collected as part of HDSS, such data were unavailable for this study. The available data only has ICD-9 (until 2001) and ICD-10 (post 2001) codes, but otherwise no specific details on identifying water-borne disease morbidity data. A medical expert's help was sought to use the ICD codes to narrow the causes potentially related to water-borne diseases. Based on this indirect method of obtaining water-borne disease morbidity data, similar analysis was performed as was done using all-cause mortality data in Figure 4e,f. The results were similar and showed no significant difference in inside/outside population. However, we note that this indirect method of obtaining water-borne disease morbidity data has its own limitation in addition to inconsistent coding schemes before and after 2001. If further health-outcome specific data were available in future, including records related to water-borne diseases or nutrition in children, we could perform a focused comparison of morbidity rates corresponding to water-borne diseases or malnutrition.

Machine-learning approaches provide further opportunities to integrate socio-economic data with other types of ancillary data, e.g., investment in public/private water infrastructure, data on fishing and farming productivity, data on loan amounts and types, hydro-climatic data, etc. Inclusion of these datasets combined with spatio-temporal analysis would constitute an interesting work to further study the embankment's impact on the environment and people. By reformulating the problem and restructuring the data, the machine-learning approaches and the framework implemented in this study can be used to answer questions relevant to related studies, e.g., identify the most informative variables indicative of owning a deep tubewell, using solar panels for electricity, contracting certain water-borne diseases, etc., which can provide valuable information to future developmental interventions.

## 6. Conclusions

Overall, the available socio-economic indicators and mortality, migration data do not provide a strong predictive value of the location of inside vs. outside residents. The study reinforces findings in the past literature around the immediate impacts of the embankment but does not find those are continued three decades later, with certain well-being indicators evening out across the two defined areas. The proposed approaches are particularly suitable in the face of large numbers of variables and samples. These methods are applicable to evaluate the environmental and socio-economic impacts of human interventions in general beyond the context of Bangladesh.

Moreover, the proposed approaches provide a new framework to identify indicators that are relevant to evaluate the impacts of interventions. Results show households inside the embankment owned a larger proportion of agricultural land within a decade of the embankment's construction. Similarly, the embankment is associated with the differential movement of people, with more people moving inward vs. outward within the study area. However, the difference appears to be evening out in recent years.

This work performs a quantitative analysis of the impacts of embankment using machine-learning approaches. By providing rigorous analytical tools, these approaches may be relevant to tackle the global challenges of flood risks. Such principled analysis is key towards evaluating both short-term and long-term as well as intended and unintended consequences of interventions, providing evidence to support future actions and policies related to combating climate change.

**Author Contributions:** Conceptualization, A.M., A.F., D.J.B., R.H.; methodology, A.M., A.F., D.J.B., R.H., D.A.C.; software, A.M.; validation, A.M., A.F., D.J.B., D.A.C.; formal analysis, A.M., A.F., D.J.B., R.H.; investigation, A.M., A.F., D.J.B., M.S., R.H.; resources, M.S.I., R.H., D.A.C.; data curation, A.M., A.F.; writing–original draft preparation, A.M., A.F., M.S.; writing–review and editing, A.M., A.F., D.J.B., M.S., M.S.I., R.H.,D.A.C.; visualization, A.M., D.J.B., M.S.; supervision, D.J.B., R.H., D.A.C.; funding acquisition, R.H., D.A.C. All authors have read and agreed to the published version of the manuscript.

**Funding:** This document is an output from the REACH program funded by UK Aid from the UK Department for International Development (DFID) for the benefit of developing countries (Aries Code 201880). However,

the views expressed and information contained in it are not necessarily those of or endorsed by DFID, which can accept no responsibility for such views or information or for any reliance placed on them.

**Acknowledgments:** The HDSS data was kindly provided by icddr,b. icddr,b acknowledges its donors who provide unrestricted support to icddr,b for its operation and research. Current donors include the Government of the People's Republic of Bangladesh, Global Affairs Canada(GAC), The Swedish International Development Cooperative Agency (Sida), and the Department of International Development (UK Aid).

**Conflicts of Interest:** The authors declare no conflict of interest. The funders had no role in the design of the study; in the collection, analyses, or interpretation of data; in the writing of the manuscript, or in the decision to publish the results.

## Abbreviations

The following abbreviations are used in this manuscript:

| | |
|---|---|
| FAP | Flood Action Plan |
| NWP | National Water Policy |
| NWMP | National Water Management Plan |
| MDIP | Meghna–Dhonagoda Irrigation Project |
| HYV | High Yielding Variety |
| HDSS | Health and Demographic Surveillance System |
| SES | Socio-Economic Survey |
| icddr,b | International Centre for Diarrheal Disease Research, Bangladesh |
| LR | Logistic Regression |
| RF | Random Forest |
| SAE | Stacked Auto-Encoders |
| PCA | Principal Component Analysis |
| GP | Gaussian Processes |
| ROC | Receiver Operating Characteristics |
| AUC | Area Under ROC Curve |

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
