# Peer review of "Machine Learning to Evaluate Impacts of Flood Protection in Bangladesh, 1983–2014"

_water, doi:10.3390/w12020483_

Round 1

Reviewer 1 Report

This is an interesting and novel approach to examining the effectiveness of flood protection infrastructure in Bangladesh.  My main comment to authors is to make clearer up front the study goal, and then in the conclusion section return to this goal and conclude clearly with respect to it, including providing a bit more discussion of the evaluation literature on this topic. Ancillary findings about the potential utility of this dataset and these methods to answering other study questions should either be left out, or made as side-comments.  These currently seem to appear as main conclusions.

Para 1.  Authors could strengthen the rationale for this study by noting that adaptation evaluation has been noted as among weakest components of the CC literature, so this paper may contribute to a key gap in knowledge.

Para 2.  Useful to explicitly state here that you are citing the past evaluation literature on Bangladesh flood infra.

Line 47.  The meaning of this summary sentence is not clear.  Perhaps it could be re-phrased.  (In sum, the Bangladesh flood protection evaluation literature says XXXX?)

Line 51.  Please define “flood control,” “flood management” and “integrated flood management” – and how this relates to “full protection” “reasonable protection” and “flood-proofing” so the reader is aware of the difference in investment/goal. 

Line 60.  Provide a bit more explanation of the Bangladesh Delta Plan 2100 and why it is relevant to your paper.  How does your chosen MDIP site related to the Delta Plan?

Before 1.1 Context and Related Works.  It is confusing that authors launch into a detailed discussion of the study site before stating the study goal.  I understand that the study is opportunistic, taking advantage of the data from the HDSS from the ICDDR,B, and it is important to lay all of this out as the authors do.  But I believe the paper will be clearer with addition of a simple, direct goal statement for your study here, before describing the study site:  i.e., something roughly like: “In order to further clarify evaluation outcomes in Bangladesh flood management, we took advantage of an existing data set to evaluate the long-term impact of flood infrastructure on a large project in Bangladesh, using machine learning.”   Otherwise, the reader does not know where you are going…

Line 101.  Please define briefly “machine learning” (I realize section 2.2 goes into detail about how machine learning has been used here, but there is no definition – helpful to have it here when first mentioned.)

Section 2.11 Socioeconomic Survey Data.  For clarity, it would be helpful to list the main variables assessed here (as you do with events data).  Particularly, it would be important to understand which of these SES variables are associated with enhanced wellbeing that could be conferred (in terms of protection) due to the embankment – housing status, agricultural assets, etc. – since that is what is really behind the research question:  Are people better off or not with this embankment protection?

Section 2.1.2 Events Data.  I am presuming mortality data is all-cause?  The mortality outcomes are key to understanding health-related/wellbeing protective capacity of the embankment (as above); but what about other health event outcomes?  E.g., does the HDSS from the ICDDR,B not have water-borne disease morbidity data?

Section 2.2 Machine Learning Approaches.  The two sets of questions are clear and robust.  

Results.  3.2  As you present it, it seems the socioeconomic data analysis suggests little detectable difference in variables inside vs outside the embankment, and that factors other than the embankment were more important to SES variables (e.g., the gasline, deep wells, other infrastructure, etc.). 

3.3 As you present it, it seems that mortality and migration outcomes were not different inside and outside embankment over time.  While there was observed difference in movement, I am not sure if I understand your hypothesis about movement with regard to the embankment.  This could be more clearly developed.

Section 3.4.  Hydroclimatic Data.  It is essential to link the analysis to actual flood events data.  As above, your analysis would suggest that at least for this one-third section of the site, the embankment was not clearly successful at protecting people from flooding.  Could this have affected peoples’ perceptions and resulting movements (see below).  It is not clear how hydroclimatic data for the study area compare to the rest of the site?  It seems from what is said that there were important differences, so this should be clearer in order to say anything about generalizability (see below).

Discussion.

Line 309.  More accurate to say the study cannot generalize due to small study size and (presumably) differences in hydroclimatic conditions.

Line 317.  As above, it would be useful to compare to morbidity due to water borne disease, does the HDSS not have some data?

The discussion section seems to lack a full comparison with the existing research on Bangladesh flood infrastructure evaluation (or at least it is not clearly stated what this literature is), and a statement regarding how these findings compare with that literature is needed.  

Conclusion.  It is important to have conclusion section.  My main suggestion would be in concluding to clearly link the study question and the main findings more clearly.  What does the study conclude about how effective the embankment was at flood protection, as judged by key SES and event data?  The ancillary benefits of the study – identifying useful variables in this dataset that could help inform other study questions – should be treated as a side outcome, not the main outcome of the study.

Could use a bit of editing for English usage, grammar, spelling in places

Author Response

We thank the reviewers for their insightful comments. We have incorporated their suggestions in the updated manuscript. For reference, the revised texts have been identified using the line numbers in the updated manuscript. Wherever applicable, we have provided additional details below to further clarify our response. For clarity, the reviewer's comments are in italics and our response in regular font style.

Comments and Suggestions for Authors

This is an interesting and novel approach to examining the effectiveness of flood protection infrastructure in Bangladesh.  My main comment to authors is to make clearer up front the study goal, and then in the conclusion section return to this goal and conclude clearly with respect to it, including providing a bit more discussion of the evaluation literature on this topic. Ancillary findings about the potential utility of this dataset and these methods to answering other study questions should either be left out, or made as side-comments.  These currently seem to appear as main conclusions.

As suggested by the reviewer, we have modified the introduction, discussion, conclusion sections. We have added the following paragraph in the introduction, which also answers few of the comments below.

Revised text: Lines 82-88 “The changes in Bangladesh Delta are affected by many factors, upstream interventions being one of them. In this work, we evaluate the impact of flood infrastructure on a large project in Bangladesh. We took advantage of an existing data set of this project in order to further clarify evaluation outcomes in Bangladesh flood management using machine learning approaches. These approaches provide robust metrics to evaluate the impacts of interventions as well as identifying the most informative indicators.”

As suggested, we have modified and separated the discussion and conclusion sections. We moved and shortened the texts related to ancillary utility of the proposed approach towards the end of the discussion section. We modified the conclusion section to highlight the findings with respect to the study goals identified in the introduction.

Para 1.  Authors could strengthen the rationale for this study by noting that adaptation evaluation has been noted as among weakest components of the CC literature, so this paper may contribute to a key gap in knowledge.

Revised text: Lines 32-36 “Although success of these interventions relies on developing principled methods to monitor and evaluate the impacts across different environmental and socioeconomic factors [Sills2017], currently there is a lack of such methods in the literature. This work aims to bridge a key gap in knowledge by proposing rigorous analytical methods to evaluate the impacts of adaptation measures.”

Para 2.  Useful to explicitly state here that you are citing the past evaluation literature on Bangladesh flood infra.

Revised text: Lines 37-39 “Bangladesh has had five decades of political and policy attention focused on implementing flood mitigation strategies, whose evaluation has been documented in the past literature on Bangladesh flood protection infrastructure.”

Line 47.  The meaning of this summary sentence is not clear.  Perhaps it could be re-phrased.  (In sum, the Bangladesh flood protection evaluation literature says XXXX?)

Revised text: Lines 52-55 “Previous evaluations of flood protection investments in Bangladesh have widely suggested that it has been difficult to attain the stated objectives of the interventions based on only technical and economic viability, but without giving due consideration to the hydromorphological features of the floodplain and the socioeconomic condition of its inhabitants.”

Line 51.  Please define “flood control,” “flood management” and “integrated flood management” – and how this relates to “full protection” “reasonable protection” and “flood-proofing” so the reader is aware of the difference in investment/goal.

Revised text: Lines 58-67 “Hence, the priority of the Government started to shift from traditional flood control to flood management towards the later stages of the Flood Action Plan (FAP) in late eighties. Here, flood control refers to the conventional method of constructing an embankment and drainage regulator whereas flood management refers to mitigating flood damage without causing degradation of the floodplain environment, which might involve implementing floodplain land use regulation that identifies floodplain zones and enforces appropriate planning and design during construction of infrastructures in these floodplain zones to account for flood and preservation of floodplain resources and environment. There was a real paradigm shift to integrated flood management, i.e. covering issues relevant to not only flood but also drainage, irrigation, navigation, environment and socioeconomic development, which was subsequently reflected in the National Water Policy (NWP) in 1999 \cite{MoWR1999} and the National Water Management Plan (NWMP) \cite{Warpo2001}.” 

Line 60.  Provide a bit more explanation of the Bangladesh Delta Plan 2100 and why it is relevant to your paper.  How does your chosen MDIP site related to the Delta Plan?

Revised text: Lines 73-81 “The Bangladesh Delta Plan 2100 also gives more emphasis to restoration, redesign and modification… In order to properly evaluate the impacts of these investments, this work strongly recommends early on adoption of principled analytical methods, considering socio-economy and environment across longer time frames, so that appropriate studies are conducted at the outset, e.g. baseline and periodic survey and monitoring to assess the impact of adaptations.”

Revised text: Lines 82-84 “The changes in Bangladesh Delta are affected by many factors, upstream interventions being one of them. In this work, we evaluate the impact of flood infrastructure on a large project in Bangladesh.”

Before 1.1 Context and Related Works.  It is confusing that authors launch into a detailed discussion of the study site before stating the study goal.  I understand that the study is opportunistic, taking advantage of the data from the HDSS from the ICDDR,B, and it is important to lay all of this out as the authors do.  But I believe the paper will be clearer with addition of a simple, direct goal statement for your study here, before describing the study site:  i.e., something roughly like: “In order to further clarify evaluation outcomes in Bangladesh flood management, we took advantage of an existing data set to evaluate the long-term impact of flood infrastructure on a large project in Bangladesh, using machine learning.”   Otherwise, the reader does not know where you are going…

Revised text: Lines 84-87 “We took advantage of an existing data set of this project in order to further clarify evaluation outcomes in Bangladesh flood management using machine learning approaches. We chose the site also because it had robust historic continuous data to trial machine learning that enabled the type of analysis that other sites could not.”

Line 101.  Please define briefly “machine learning” (I realize section 2.2 goes into detail about how machine learning has been used here, but there is no definition – helpful to have it here when first mentioned.)

Revised text: Lines 84-88 “We took advantage of … using machine learning approaches… These approaches provide robust metrics to evaluate the impacts of interventions as well as identifying the most informative indicators.”

Section 2.11 Socioeconomic Survey Data.  For clarity, it would be helpful to list the main variables assessed here (as you do with events data).  Particularly, it would be important to understand which of these SES variables are associated with enhanced wellbeing that could be conferred (in terms of protection) due to the embankment – housing status, agricultural assets, etc. – since that is what is really behind the research question:  Are people better off or not with this embankment protection?

As suggested by the reviewer we have made the following modifications in the manuscript. In addition, Section 3.2 highlights the specific SES variables that are determined to be important indicators of difference between inside/outside population.

Revised text: Lines 166-171 “Considering 1982 as a baseline pre-embankment period resulted in only four variables (agricultural land ownership, primary drinking water source, number of cow/buffaloes/goats owned, and boat ownership) and twelve variables (previous variables plus homestead land ownership, household assets – sofa, chair/table, showcase, radio, TV, bike/bicycle, primary rood structure, and sanitation facility type) equivalent across all four and the later three years respectively.”

Section 2.1.2 Events Data.  I am presuming mortality data is all-cause?  The mortality outcomes are key to understanding health-related/wellbeing protective capacity of the embankment (as above); but what about other health event outcomes?  E.g., does the HDSS from the ICDDR,B not have water-borne disease morbidity data?

Yes, the mortality data being modelled is all-cause. Although other health event outcome related data were collected as part of HDSS, such data was unavailable for this study. The available data only has ICD-9 (until 2001) and ICD-10 (post 2001) codes, but otherwise no specific details on identifying water borne disease morbidity data. A medical expert’s help was sought to use the ICD codes to narrow the causes potentially related to water-borne diseases. Based on this indirect method of obtaining water borne disease morbidity data, similar analysis was performed as done using all-cause morbidity data in Figures 4e and 4f (previously 3e and 3f). The results were similar and showed no significant difference in inside/outside population. Since this indirect method of obtaining water borne disease morbidity data has its own limitation in addition to inconsistent coding schemes pre/post 2001, we decided to not show these results.

We mention in lines 376-378 “If further health-outcome specific data were available in future, e.g. records related to water-borne diseases or nutrition in children, we could perform a focused comparison of mortality rates corresponding to water-borne diseases or malnutrition.”

Section 2.2 Machine Learning Approaches.  The two sets of questions are clear and robust.  

Results.  3.2  As you present it, it seems the socioeconomic data analysis suggests little detectable difference in variables inside vs outside the embankment, and that factors other than the embankment were more important to SES variables (e.g., the gasline, deep wells, other infrastructure, etc.). 

Yes, that is correct.

3.3 As you present it, it seems that mortality and migration outcomes were not different inside and outside embankment over time.  While there was observed difference in movement, I am not sure if I understand your hypothesis about movement with regard to the embankment.  This could be more clearly developed.

Revised text: Lines 181-183 “We hypothesize that if the flood-protected area were to provide increased socioeconomic benefits and stability, people would more likely move to inside the flood-protected area.”

Section 3.4.  Hydroclimatic Data.  It is essential to link the analysis to actual flood events data.  As above, your analysis would suggest that at least for this one-third section of the site, the embankment was not clearly successful at protecting people from flooding.  Could this have affected peoples’ perceptions and resulting movements (see below).  It is not clear how hydroclimatic data for the study area compare to the rest of the site?  It seems from what is said that there were important differences, so this should be clearer in order to say anything about generalizability (see below).

Please note that the hydroclimatic data, i.e. Figures 5a, 5c, and 5d (previously 4a, 4c, and 4d), correspond to the entire study site but the events data (including the movement data) correspond to the one-third of the study site geographically, which does have implications in terms of generalizing the results. Texts modified accordingly below in response to the latter comment.

The reviewer’s observation (regarding how the embankment’s failure may have affected peoples’ perceptions and resulting movements) is plausible and one that we also made. However, given the available data, although we can offer plausible explanation, we are unable to make conclusive statements. We mention in lines 364-365 “Further analysis of relevant data (e.g. financial loss due to floods) may allow us to study the relationship between hydro-climatic events and socioeconomic behaviour.”

Discussion.

Line 309.  More accurate to say the study cannot generalize due to small study size and (presumably) differences in hydroclimatic conditions.

Revised text: Lines 369-370 “The study may not generalize due to small study size and differences in hydroclimatic conditions.”

Line 317.  As above, it would be useful to compare to morbidity due to water borne disease, does the HDSS not have some data?

Please refer to our response above.

The discussion section seems to lack a full comparison with the existing research on Bangladesh flood infrastructure evaluation (or at least it is not clearly stated what this literature is), and a statement regarding how these findings compare with that literature is needed.  

As suggested, we have restructured and modified the discussion and conclusion sections. We have also added a paragraph on the existing research on Bangladesh flood infrastructure evaluation, specifically using the HDSS data. This paragraph was more appropriately placed in Section 1.1 Context and Related Work

Revised text: Lines 119-149 “Earlier studies… ability to include in longer-term impact assessments.”

Referring to the previous studies on the evaluation of the impact of embankment, we have also modified the following sentence in the conclusion section.

Revised text: Lines 393-396 “The study reinforces findings in the past literature around the immediate impacts of the embankment but does not find those are continued three decades later, with certain well-being indicators evening out across the two defined areas.”

Conclusion.  It is important to have conclusion section.  My main suggestion would be in concluding to clearly link the study question and the main findings more clearly.  What does the study conclude about how effective the embankment was at flood protection, as judged by key SES and event data?  The ancillary benefits of the study – identifying useful variables in this dataset that could help inform other study questions – should be treated as a side outcome, not the main outcome of the study.

As suggested, we have restructured and modified the conclusion section. We have highlighted the primary findings in the first paragraph while shortening and shifting the secondary findings in the second paragraph.

Revised text: Lines 392-403 “Overall, the available socioeconomic… in recent years.”

Could use a bit of editing for English usage, grammar, spelling in places

Proof-read again by the authors.

Reviewer 2 Report

The paper aims to use machine learning to study the long-term impacts of flood protection in Bangladesh. Available data include socioeconomic survey and past events data about death, migration, etc. Specifically, it was tested test whether the embankment has affected the welfare of people over time, benefiting those living inside more than those living outside.

The topic is of interest for Water and a wide audience.

My main concern regards the methodology: as far as I know, machine learning needs two sets of data – one for “training” and one for the analysis. In the paper one dataset was presented only.

The second concern regards the structure. The Introduction sections mainly reports about Bangladesh instead of giving an overall review of the paper, introducing the and setting it in a broader context, gradually narrowing the topic down to a research problem.

A better structure would be:

Introduction: description of the problem in a wider context narrowing down to a specific (and generally worldwide applicable) problem Literature review: critical review of existing work and methodologies, particularly identifying gaps and priorities. Methodology: the adopted methodology. Case Study: LL33-103, summarised plus datasets; I would say that also LL191-202 belong here. Input data should follow the description of the methods. Results. Discussion. Conclusion (missing at the moment, besides LL345-349).

I am listing below some additional notes.

- is the study to evaluate impacts of flood protection (title) or climate change adaptation (L1)? ;

- “adaptation” is mainly used singular, on the contrary “strategies of adaptation should be used;

- “long-term as “short-term” needs “-“ between the two words;

- Fig. 1g is not readable;

- a table of all the considered variables (with description) is needed;

- Fig. 2i to 2viii are too small;

- “Not shown” results (LL 212,276,277) shall either not mentioned or shown;

- Sec. 3.4 seems a stand-alone section, not connected with the machine learning techniques or the previous analysis;

- the model has not been verified.

Author Response

We thank the reviewers for their insightful comments. We have incorporated their suggestions in the updated manuscript. For reference, the revised texts have been identified using the line numbers in the updated manuscript. Wherever applicable, we have provided additional details below to further clarify our response. For clarity, the reviewer's comments are in italics and our response in regular font style.

Comments and Suggestions for Authors

The paper aims to use machine learning to study the long-term impacts of flood protection in Bangladesh. Available data include socioeconomic survey and past events data about death, migration, etc. Specifically, it was tested test whether the embankment has affected the welfare of people over time, benefiting those living inside more than those living outside.

The topic is of interest for Water and a wide audience.

My main concern regards the methodology: as far as I know, machine learning needs two sets of data – one for “training” and one for the analysis. In the paper one dataset was presented only.

Yes, we agree with the reviewer that a machine learning approach is generally evaluated by splitting the data into two disjoint training and test sets, which is exactly what we have done while analysing the socioeconomic survey data as part of the k-fold cross-validation technique described in Section 3.1. This is a standard technique for evaluating a classification approach by repeating the training and testing process multiple times allowing us to evaluate the classifier on the entire dataset. The ROC curves and the AUCs metrics reported are based on the test data sets.

Please note that while analysing the events data, as described in Section 3.1, the Gaussian Processes were used to perform Bayesian statistical significance tests to query whether two time series data are different. Hence it would not make sense to divide the events data into training/test sets.

The second concern regards the structure. The Introduction sections mainly reports about Bangladesh instead of giving an overall review of the paper, introducing the and setting it in a broader context, gradually narrowing the topic down to a research problem.

A better structure would be:

Introduction: description of the problem in a wider context narrowing down to a specific (and generally worldwide applicable) problem Literature review: critical review of existing work and methodologies, particularly identifying gaps and priorities. Methodology: the adopted methodology. Case Study: LL33-103, summarised plus datasets; I would say that also LL191-202 belong here. Input data should follow the description of the methods. Results. Discussion. Conclusion (missing at the moment, besides LL345-349).

We have followed the MDPI journal instructions when structuring and formatting the manuscript, which consists of Introduction, Materials and Methods, Results, Discussion, and (optional) Conclusion. As suggested by the reviewer, we have restructured and modified the manuscript wherever applicable, including modifying the introduction, adding more literature review specific to the study area, and a separate Conclusion section. Please note we chose to describe the dataset prior to the methods because the description of the dataset helps motivate the machine learning approaches implemented in this work, i.e. classification approaches to model socioeconomic survey data and Gaussian processes to model events data.

Revised text (Our contribution): Lines 32-36 “Although success of these interventions relies on developing principled methods to monitor and evaluate the impacts across different environmental and socioeconomic factors [Sills2017], currently there is a lack of such methods in the literature. This work aims to bridge a key gap in knowledge by proposing rigorous analytical methods to evaluate the impacts of adaptation measures.”

Revised text (Study outline): Lines 82-88 “The changes in Bangladesh Delta are affected by many factors, upstream interventions being one of them. In this work, we evaluate the impact of flood infrastructure on a large project in Bangladesh. We took advantage of an existing data set of this project in order to further clarify evaluation outcomes in Bangladesh flood management using machine learning approaches. These approaches provide robust metrics to evaluate the impacts of interventions a well as identifying the most informative indicators.”

Revised text (Additional literature review): Lines 119-149 “Earlier studies… impact assessments.”

Revised text (Discussion section): Lines 367-390 “The study has limitations… future developmental interventions.”

Revised text (Conclusion section): Lines 392-408 “Overall, the available socioeconomic… combating climate change.”

I am listing below some additional notes.

- is the study to evaluate impacts of flood protection (title) or climate change adaptation (L1)? ;

The primary study goal is to evaluate the impacts of flood protection. The proposed framework is equally applicable to evaluate the impacts of any adaptation strategies in general. As the reviewer suggests starting with the broader picture and narrowing down to the specific problem, we begin with the broader challenge, which is a gap in knowledge in terms of evaluating the impacts of climate change adaption strategies in general, followed by a specific example of such adaptation measures, flood protection. As the literature suggests, increased investments related to climate change adaptation measures are expected in future, flood protection infrastructure to mitigate increased expected flooding being one of those investments.

- “adaptation” is mainly used singular, on the contrary “strategies of adaptation should be used;

“adaptation” modified to “adaptation strategies” , “adaptation measures”, or “adaptation action and policies”.

- “long-term as “short-term” needs “-“ between the two words;

Texts have been modified appropriately.

- Fig. 1g is not readable;

Fig. 2a (previously Fig. 1a) was enlarged and moved earlier in the manuscript to Fig. 1, which provides a visual reference to Fig. 2g (previously Fig. 1g). Please note the primary purpose of Fig. 2g is to only show that the cluster of households identified by the classifier lies close to the icddr,b hospital, which is described in Lines 301-312.

- a table of all the considered variables (with description) is needed;

As suggested by the reviewer we have made the following modifications in the manuscript. In addition, Section 3.2 highlights the specific socioeconomic variables that are determined to be important indicators of difference between inside/outside population.

Revised text: Lines 166-171 “Considering 1982 as a baseline pre-embankment period resulted in only four variables (agricultural land ownership, primary drinking water source, number of cow/buffaloes/goats owned, and boat ownership) and twelve variables (previous variables plus homestead land ownership, household assets - sofa, chair/table, showcase, radio, TV, bike/bicycle, primary rood structure, and sanitation facility type) equivalent across all four and the later three years respectively.”

- Fig. 2i to 2viii are too small;

Fig. 3a (previously Fig. 2a) was enlarged and moved earlier in the manuscript to allow Fig. 3i-3vii (previously Fig. 2i-2vii) to be bigger.

- “Not shown” results (LL 212,276,277) shall either not mentioned or shown;

The plots were not shown because they look very similar to the ones shown in the manuscript and does not add any further utility to the description in the text. Although the plots were not shown, we feel it is important to report those analyses were performed for completeness because the texts answer the obvious follow-up questions succinctly.

- Sec. 3.4 seems a stand-alone section, not connected with the machine learning techniques or the previous analysis;

We agree with the reviewer that Section 3.4 does not use any machine learning approach. The primary purpose of this section was to highlight the need to integrate hydroclimatic data with socioeconomic and demographic data in order to fully understand the impact of embankment. The qualitative comparison suggest utility in such integration using machine learning approaches should relevant dataset be available in future, as mentioned in lines 364-365.

- the model has not been verified.

As described in Section 3.1, the machine learning approaches were rigorously evaluated using standard techniques using k-fold cross-validation for classification approaches and Bayesian statistical significance tests for regression approach. We have reported both qualitative and quantitative results in the manuscript.

Round 2

Reviewer 1 Report

I have reviewed the authors' revisions to the MS.  I believe the authors have substantially taken account of the issues I raised in my initial review.  I appreciate in particular the efforts to make the study goal and conclusions clearer and more mutually coherent, as well as the helpful definitions and context on Bangladesh's current flood planning and the various flood defense definitions.

There are three places where I believe minor further clarification would be helpful.  

(1) I had suggested "machine learning" be defined when it is first mentioned in the MS.  The authors have amended language to explain why machine learning can be useful to evaluate this study question.  But they have not provided a definition of machine learning.  What I mean is something like this which comes from Wikipedia: "ML is a sub-set of artificial intelligence that uses algorithms and statistical models without specific instructions but rather pattern recognition..." etc.  I am sure you can improve upon this.  Please provide a simple definition.  Not all journal readers know what machine learning is.

(2) My review had raised the issue of the potential for water-borne disease mortality to serve as an additional outcome.  The authors' response is quite helpful and demonstrates effort to make the most of the data.  While I fully concur with the authors regarding the limitations, as they present them, of the water-borne disease morbidity data as a main analysis, my suggestion would be to briefly outline these findings in the Discussion (not Results) section as an "additional analysis."  The advantage of doing this is that it does indeed show you tried to make best use of your available data; that your water-borne disease findings -- even with their robustness drawbacks -- support the main analysis findings which is helpful to your conclusions; and finally, inclusion allows you to lay a pathway for future work, i.e., "a future analysis could more fully evaluate the water-borne disease outcomes" etc.  This is up to you, but could be helpful to the paper if you've already done the work.

(3)  The revised text on the "4 plus 12" SES variables is still not clear.  I realize that when one is writing up a study it all seems very evident to the study team. But it's good to remember that people reading your paper do not have your intimacy with the study/data -- and also have very little time to figure things out.  So if you want them to "get it," try to be as clear as possible!

Nice and helpful study -- good luck with this interesting work!

Author Response

We thank the reviewers for their insightful comments. We have incorporated their suggestions in the updated manuscript. For reference, the revised texts have been identified using the line numbers in the updated manuscript. Wherever applicable, we have provided additional details below to further clarify our response.

Comments and Suggestions for Authors

I have reviewed the authors' revisions to the MS.  I believe the authors have substantially taken account of the issues I raised in my initial review.  I appreciate in particular the efforts to make the study goal and conclusions clearer and more mutually coherent, as well as the helpful definitions and context on Bangladesh's current flood planning and the various flood defense definitions.

There are three places where I believe minor further clarification would be helpful.  

(1) I had suggested "machine learning" be defined when it is first mentioned in the MS.  The authors have amended language to explain why machine learning can be useful to evaluate this study question.  But they have not provided a definition of machine learning.  What I mean is something like this which comes from Wikipedia: "ML is a sub-set of artificial intelligence that uses algorithms and statistical models without specific instructions but rather pattern recognition..." etc.  I am sure you can improve upon this.  Please provide a simple definition.  Not all journal readers know what machine learning is.     

Revised text: Lines 85-89 “Machine learning, generally considered a subset of artificial intelligence, is a field of study that uses algorithms and statistical models to learn patterns from data so that useful inference may be made about new data. Machine learning approaches are useful in the context of this study because they provide robust metrics to evaluate the impacts of interventions as well as identifying the most informative indicators.”

(2) My review had raised the issue of the potential for water-borne disease mortality to serve as an additional outcome.  The authors' response is quite helpful and demonstrates effort to make the most of the data.  While I fully concur with the authors regarding the limitations, as they present them, of the water-borne disease morbidity data as a main analysis, my suggestion would be to briefly outline these findings in the Discussion (not Results) section as an "additional analysis."  The advantage of doing this is that it does indeed show you tried to make best use of your available data; that your water-borne disease findings -- even with their robustness drawbacks -- support the main analysis findings which is helpful to your conclusions; and finally, inclusion allows you to lay a pathway for future work, i.e., "a future analysis could more fully evaluate the water-borne disease outcomes" etc.  This is up to you, but could be helpful to the paper if you've already done the work.Revised text: Lines 37-39 “Bangladesh has had five decades of political and policy attention focused on implementing flood mitigation strategies, whose evaluation has been documented in the past literature on Bangladesh flood protection infrastructure.”

Revised text in Discussion section: Lines 381-392 “Although other health event outcome related data were collected as part of HDSS, such data were unavailable for this study. The available data only has ICD-9 (until 2001) and ICD-10 (post 2001) codes, but otherwise no specific details on identifying water-borne disease morbidity data. A medical expert's help was sought to use the ICD codes to narrow the causes potentially related to water-borne diseases. Based on this indirect method of obtaining water borne disease morbidity data, similar analysis was performed as was done using all-cause mortality data in Figs. 4e and 4f. The results were similar and showed no significant difference in inside/outside population. However, we note that this indirect method of obtaining water-borne disease morbidity data has its own limitation in addition to inconsistent coding schemes before and after 2001. If further health-outcome specific data were available in future, including records related to water-borne diseases or nutrition in children, we could perform a focused comparison of morbidity rates corresponding to water-borne diseases or malnutrition.”

(3)  The revised text on the "4 plus 12" SES variables is still not clear.  I realize that when one is writing up a study it all seems very evident to the study team. But it's good to remember that people reading your paper do not have your intimacy with the study/data -- and also have very little time to figure things out.  So if you want them to "get it," try to be as clear as possible!

Revised text: Lines 239-246 “Considering 1982 as a baseline pre-embankment period resulted in only four variables equivalent across all four years. Those variables were agricultural land ownership, primary drinking water source, number of cow/buffaloes/goats owned, and boat ownership. Considering only the later three years (1996, 2005, and 2014) resulted in slightly more (twelve) variables equivalent across all three years. Those variables were agricultural land ownership, homestead land ownership, primary drinking water source, number of cow/buffaloes/goats owned, boat ownership, household assets - sofa, chair/table, showcase, radio, TV, bike/bicycle, primary rood structure, and sanitation facility type.” 

Reviewer 2 Report

The structure is still not working well. The Method sections should start with the methodology, which should include also the methodological description of Sec. 3 (e.g. about ROC curve); input data should be included in a section "Case Study" (after Method) which should include also the part of the Introduction related to the case study area.

All references appear with "?".

Author Response

We thank the reviewers for their insightful comments. We have incorporated their suggestions in the updated manuscript. For reference, the revised texts have been identified using the line numbers in the updated manuscript. Wherever applicable, we have provided additional details below to further clarify our response.

Comments and Suggestions for Authors

The structure is still not working well. The Method sections should start with the methodology, which should include also the methodological description of Sec. 3 (e.g. about ROC curve); input data should be included in a section "Case Study" (after Method) which should include also the part of the Introduction related to the case study area.

As suggested by the reviewer, we have revised the structure of the manuscript. While doing so, we have tried our best to accommodate both reviewers’ suggestions. The other reviewer has recommended providing Bangladesh-specific details early on as part of the introduction section. We believe the reviewer is referring to this part of the introduction as “case study”. Other than that, as advised by the reviewer, we have modified the structure corresponding to methods, evaluation metric (ROCs, etc.), and datasets. We hope the reviewer is satisfied with the revised structure below:

Section 1 Introduction

Section 1.1 Context and Related Works

Section 2 Methods

Section 2.1 Machine learning approaches

Section 2.2 Evaluation metric

Section 3 Data

Section 3.1 Socioeconomic survey data

Section 3.2 Events data

Section 4 Results

Section 4.1 Socioeconomic survey data analysis

Section 4.2 Events data analysis

Section 4.3 Hydro-climatic data analysis

Section 5 Discussion

Section 6 Conclusions

All references appear with "?".

We apologize for this latex-related error. All references now appear correctly without “?” signs.